# External Validation of the Dutch SOURCE Survival Prediction Model in Belgian Metastatic Oesophageal and Gastric Cancer Patients

**DOI:** 10.3390/cancers12040834

**Published:** 2020-03-31

**Authors:** J.J. van Kleef, H.G. van den Boorn, R.H.A. Verhoeven, K. Vanschoenbeek, A. Abu-Hanna, A.H. Zwinderman, M.A.G. Sprangers, M.G.H. van Oijen, H. De Schutter, H.W.M. van Laarhoven

**Affiliations:** 1Cancer Center Amsterdam, Amsterdam University Medical Centers, University of Amsterdam, Department of Medical Oncology, 1105 AZ Amsterdam, The Netherlands; j.j.vankleef@amsterdamumc.nl (J.J.v.K.); h.g.vandenboorn@amsterdamumc.nl (H.G.v.d.B.); m.g.vanoijen@amsterdamumc.nl (M.G.H.v.O.); 2Department of Research & Development, Netherlands Comprehensive Cancer Organisation (IKNL), 3511 DT Utrecht, The Netherlands; R.Verhoeven@iknl.nl; 3Belgian Cancer Registry, 1210 Brussels, Belgium; katrijn.vanschoenbeek@kankerregister.org (K.V.); Harlinde.DeSchutter@kankerregister.org (H.D.S.); 4Department of Medical Informatics, Amsterdam University Medical Centers, University of Amsterdam, 1105 AZ, Amsterdam, The Netherlands; a.abu-hanna@amsterdamumc.nl; 5Department of Clinical Epidemiology, Biostatistics and Bioinformatics, Amsterdam University Medical Centers, University of Amsterdam, 1105 AZ Amsterdam, The Netherlands; a.h.zwinderman@amsterdamumc.nl; 6Department of Medical Psychology, Amsterdam University Medical Centers, Academic Medical Center, University of Amsterdam, Amsterdam Public Health Research Institute, 1105 AZ Amsterdam, The Netherlands; m.a.sprangers@amsterdamumc.nl

**Keywords:** prediction model, external validation, oesophageal cancer, gastric cancer

## Abstract

The SOURCE prediction model predicts individualised survival conditional on various treatments for patients with metastatic oesophageal or gastric cancer. The aim of this study was to validate SOURCE in an external cohort from the Belgian Cancer Registry. Data of Belgian patients diagnosed with metastatic disease between 2004 and 2014 were extracted (*n* = 4097). Model calibration and discrimination (c-indices) were determined. A total of 2514 patients with oesophageal cancer and 1583 patients with gastric cancer with a median survival of 7.7 and 5.4 months, respectively, were included. The oesophageal cancer model showed poor calibration (intercept: 0.30, slope: 0.42) with an absolute mean prediction error of 14.6%. The mean difference between predicted and observed survival was −2.6%. The concordance index (c-index) of the oesophageal model was 0.64. The gastric cancer model showed good calibration (intercept: 0.02, slope: 0.91) with an absolute mean prediction error of 2.5%. The mean difference between predicted and observed survival was 2.0%. The c-index of the gastric cancer model was 0.66. The SOURCE gastric cancer model was well calibrated and had a similar performance in the Belgian cohort compared with the Dutch internal validation. However, the oesophageal cancer model had not. Our findings underscore the importance of evaluating the performance of prediction models in other populations.

## 1. Introduction

Oesophagogastric cancer has a dismal prognosis. Patients diagnosed with metastatic disease face a median overall survival (OS) time of three to five months with best supportive care (BSC) [1,2]. Survival is dependent on various prognostic factors and treatment type [3].

Patients with a relatively good Eastern Cooperative Oncology Group Performance Status (PS) of 0–2, may be eligible for chemotherapy, targeted therapy, or even palliative surgery [4,5].

Brachytherapy, external radiotherapy, or stent placement may be deployed to relieve symptoms, such as dysphagia, and/or to reduce tumour growth [6,7].

Palliative treatments often have uncertain and limited benefit while the treatment burden can be high. Ideally, shared decision-making should be applied where patient preferences and values are taken into account during decision making [8]. Accurate and balanced information about treatment options tailored to the individual patient should be provided. However, oncologists were found to rarely discuss the potential pros and cons of palliative treatment and the BSC option [9,10,11,12]. This may, at least in part, be due to the complexity of predicting outcomes for individual patients [13].

Prediction models can aid such individual risk estimation. Additionally, they can help quantify risks and benefits in an understandable manner to patients which allows them to more actively participate in the decision-making process [14,15]. Such prediction models will only live up to their potential if they have the required model performance qualities. A recent review investigated published risk prediction models regarding oesophagogastric cancer and concluded that model performance is often poorly described and external validation limited [16]. In addition, no models in the metastatic setting were of sufficient quality for use in clinical practice.

We therefore developed the SOURCE model (stimulating evidence-based, personalised and tailored information provision to improve decision-making after oesophageal-gastric cancer diagnosis) [17]. The model makes OS predictions based on prognostic factors for metastatic oesophagogastric cancer patients. The SOURCE model was developed on a nationwide Dutch population-based cohort selected from the Netherlands Cancer Registry. Predictions regarding OS are conditional on various treatment types. Details on the input parameters, development and internal validation of the model were previously published [17].

External validation is needed to investigate the performance of the original Dutch model and to justify its use for other populations. The Belgian population was selected, because the neighbouring countries have an extensive population-based national cancer registry. Therefore, the aim of this study was to validate the SOURCE model on an external population-based cohort selected from the Belgian Cancer Registry (BCR).

## 2. Results

Overall, 4097 patients diagnosed between 2004 and 2014 registered by the BCR were included. Figure 1 depicts the selection process stratified by oesophageal and gastric cancer patients.

### 2.1. Oesophageal Cancer Patients

In total, 2514 oesophageal cancer patients were analysed of whom 97.1% died during follow-up. Most patients were male (80.8%), had a PS of 1 (65.0%) and were diagnosed with adenocarcinoma (67.3%). The median observed OS was 7.7 months. An overview of patient, tumour and treatment characteristics is given in Table 1.

Compared to the Dutch SOURCE population, the median OS time was higher for Belgian patients (7.7 vs. 5.1 months, *p* < 0.0001), see Table 1. cT3 tumours were more frequently observed in Belgian patients (45.5% vs. 22.7%) whereas the Dutch population had a cTX status in 49.9% of patients. Squamous cell carcinoma was compared to adenocarcinoma more frequently diagnosed in Belgium than in the Netherlands. Topography was not further specified in 33.1% of Belgian patients. Half of the Belgian patients were treated with chemotherapy, 10.6% received BSC and 5.8% received radiotherapy. Dutch patients received less treatment; 27.7% received chemotherapy, 26.6% BSC and 26% radiotherapy. 

### 2.2. SOURCE Oesophageal Cancer Model Validation 

Model discrimination for the oesophageal cancer population amounted to a c-index of 0.64 (0.63–0.66), see Table 2. Model calibration at six months for the overall oesophageal cancer population corresponded to an intercept and calibration slope of 0.30 (0.28–0.31) and 0.42 (0.39–0.45), respectively. The mean difference between predicted and observed survival was −2.6% with a mean absolute prediction error of 14.6% (Table 2). The corresponding calibration plot (Figure 2) shows an underestimation of OS for patients with a predicted six-month OS of ≤46% with the most prominent deviations in the lowest tertile of the plot. Overestimation of six-month OS was present for patients with a relatively good prognosis, with larger deviations on the higher end of the scale (60–80%), see Figure 2.

Figure 3 depicts mean differences between predicted and observed survival for various patient subgroups. The majority of patient subgroups (57%) showed larger differences between predicted and observed survival compared to the overall population (−2.6%).

### 2.3. Gastric Cancer Patients

In total, 1583 patients with gastric cancer were analysed of whom 98.0% died during follow-up. Details of patient, tumour and treatment characteristics are given in Table 3. More than half of the patients were male (59.8%) and had a PS of 1 (59.6%). The median observed OS was 5.4 months.

Compared to the original Dutch SOURCE population, the median OS time was longer for Belgian patients (5.4 vs. 3.9 months, *p* < 0.0001), see Table 3. The primary tumour location was not further specified in 60.1% of Belgian patients versus 8.4% of Dutch patients, and topography was assessed as an overlapping lesion in 0.5% of Belgian versus 34.5% of Dutch patients. Half (52.1%) of the Belgian patients were treated with chemotherapy versus 34.6% of Dutch patients, and 33.5% of Belgian patients received BSC versus 47.6% of Dutch patients. Detailed information regarding the location of metastases was missing in 658 (41.6%) Belgian patients.

### 2.4. SOURCE Gastric Cancer Model Validation

Model discrimination amounted to a c-index of 0.66 (0.64–0.68). Model calibration at six months for the overall gastric cancer population corresponded to an intercept and calibration slope of 0.02 (0.02–0.02) and 0.91 (0.90–0.91), respectively. The mean difference between predicted and observed survival was 2.0% with a mean absolute prediction error of 2.5% (Table 2). The corresponding calibration plot showed good calibration with no differences greater than 5% between predicted and observed survival along all prediction estimates, see Figure 2.

Differences between predicted and observed survival were greatest in terms of overestimation for patients aged 80–89 (+6.1%), and with a PS score of 3 (+5.8%) and a cN3 status (+5.5%). The majority of patient subgroups (59%) showed similar or smaller differences between predicted and observed OS compared to the overall cohort (−2.0%), see Figure 4.

## 3. Discussion

External validation of prediction models is essential for use in clinical practice [18]. This external validation study of the Dutch SOURCE model demonstrated that the oesophageal model had low transportability to the Belgian population, given its poor calibration and c-index. However, the gastric cancer model transported adequately.

The original development report of SOURCE noted a calibration slope of one and an intercept of zero for both models during internal validation. C-indices were 0.71 and 0.68 for the oesophageal and gastric cancer model, respectively [17]. In this external validation study, we did not expect a superior performance compared to the internal validation, given the different nationality and healthcare settings. Our results showed that the oesophageal model performed poorer in the Belgian population regarding its calibration and a c-index of 0.64, but the performance of the gastric cancer model was close to the original internal validation with a good calibration and a c-index of 0.66.

Calibration of the gastric cancer model showed that differences between predicted and observed OS for the entire cohort were no greater than 5% along the calibration line, indicating a well calibrated model. Differences between predicted and observed OS were small (<5%) for most patient subgroups. Older patients aged 80–90 had the largest difference (+6.1%), which still was interpreted as fair by us.

C-indices were relatively low according to our classification, indicating that the models had difficulties in making higher prediction estimates for patients who actually survived longer versus patients who had a shorter lifespan. Since the calibration was good for the gastric cancer population and the variation between prediction estimates was small, one might argue that the model had difficulties in ranking patients’ OS. 

The poor fit of the oesophageal cancer model might be explained by overfitting during model development. The oesophageal cancer model has more input parameters and interaction terms compared to the gastric cancer model (see Appendix A). Such complex models with a high number of parameters might lead to good fit for the sample population—in this case the Dutch—but predictions might not generalize to new subjects outside the sample, such as the Belgians [19].

Missing data in the Belgian cohort might be another explanation for the poor fit. In this study, >40% of data regarding the location of metastases was missing and therefore multiply imputed to avoid selection bias. This, however, is always suboptimal in comparison to having observed values. The oesophageal model compared to the gastric model contains more input parameters regarding the location of metastases (see Appendix A). Therefore, the oesophageal model validation was more subject to multiple imputation and thus uncertainty, which might explain the poorer fit.

Furthermore, adenocarcinoma and squamous cell carcinoma were combined into the same oesophageal cancer model, despite their differential biological features. Although the oesophageal cancer model contained histology as an input parameter, it is unclear to what extent this combination contributed to the poor model fit. Patient subgroup analysis showed that mean differences between predicted and observed survival for adenocarcinoma, squamous cell carcinoma, and the entire cohort were −2.9%, +1.7% and −2.6%, respectively. These mean differences did not substantially differ (see Figure 3). For the re-estimated model based on BCR data and its calibration and discrimination, see Appendix A.

### Differences between Development and Validation Datasets

Several differences in patient, tumour and treatment characteristics were observed between the Dutch and Belgian population. These include topography, cT-category and tumour differentiation grade, which might be due to missing data and/or differing cancer registration policies. In the Netherlands, data managers are centrally trained to interpret and register data in a standardised fashion. In Belgium, data collection is decentralised where clinical and pathological data is obtained by oncological care programmes and laboratories [20]. Albeit training of data managers and data cleaning is performed according to specific guidelines, differences in registration might thus be due to varying registration practices and/or interpretations [21]. In addition, BCR data regarding treatment types have been sufficiently validated. Data regarding the location of metastases were derived from Belgian hospital discharge data. This is the first study to use discharge records for this this purpose. It is, however, unknown to what extent this data might deviate from patients’ medical records. 

Taking patient selection into account, the proportion of patients with cM1 tumours at diagnosis in the BCR was substantially lower compared to the Netherlands (22.1% vs. 40.1%). Additionally, the proportion of Belgian patients with a cTXNXMX status was considerably higher (28.6% vs. 1.9%) (personal communication, 29 May 2019). So, one might argue that this Belgian cTXNXMX patient group is quite heterogeneous and that a portion of these patients had true cM1 tumours at diagnosis. These patients, however, were not included in our analysis due to lack of detail in the clinical TNM classification. It might be the case that including these patients affects case mix and survival, which could lead to a dataset more similar to the Dutch.

When looking at the use of treatment modalities in the Belgian sample, the Belgians administered chemotherapy more frequently than the Dutch. Dutch oncologists more frequently offered BSC and radiotherapy, a more conservative approach that may explain the shorter median survival [20,21,22].

Lastly, SOURCE was developed to aid decision-making between BSC and (some form of) active treatment. During model development, 26.6% (*n* = 2131) and 47.6% (*n* = 2266) of Dutch oesophageal and gastric cancer patients received BSC (no treatment). Although this relatively large cohort could aid survival estimation on BSC, it should be pointed out that these estimates may have an inherent selection bias. Patients who received BSC most likely had worse PS scores or comorbidities compared to patients who did undergo treatment. Therefore, survival estimates for a relatively fit patient considering BSC may be underestimated. Although this effect could be partially corrected by other input parameters in the model, there may still be bias in the survival predictions [17].

## 4. Materials and Methods

This manuscript was written in accordance with the TRIPOD statement [23].

The SOURCE model aims to stimulate evidence based, personalized and tailored information provision to improve decision-making after oesophageal–gastric cancer diagnosis. The model predicts overall survival for patients with metastatic oesophageal or gastric carcinoma (cM1), who did not die within 14 days after diagnosis. Patients with only distant metastases located in the head or neck region fall outside the target population of SOURCE.

Input parameters of the model include: Age, cT-category, cN-category, tumour differentiation grade, number of metastatic sites, distant lymph node metastasis only, intra-thoracic and intra-abdominal lymph node metastasis and initial treatment. The gastric cancer model also includes gender as an input parameter and the oesophageal cancer model also includes peritoneal, liver and head and neck metastases, morphology and topography. Input parameters were measured at diagnosis, before the start of treatment.

SOURCE is integrated into a web-interface and will be made freely available after extensive assessments in clinical practice. Physicians can use the model together with patients during the clinical consultation. Since medical terminology is present in the web-interface, it is recommended that physicians discuss the results from the model with the patient, in a way that is tailored to the patient’s level of understanding. It should be noted that SOURCE is developed to be a decision-aid to stimulate shared and informed decision-making. It should not and cannot replace the expertise and clinical judgement of physicians.

### 4.1. Data Source

The BCR covers more than 95% of the Belgian cancer population [24]. Patient and tumour characteristics were collected from the standard cancer registration database, which relies on notifications from both the clinical (oncology care programmes) and pathological (laboratories for pathological anatomy) network. Data regarding treatment were derived from reimbursement claims of health insurance companies. A detailed description of the BCR data and data sources is given in the Appendix A. The use of BCR data for scientific purposes is regulated by Belgian law, excluding the need for written informed consent for this study.

### 4.2. Patients

All patients diagnosed between 2004 and 2014 with a primary tumour in the oesophagus/gastroesophageal junction or stomach (ICD-10: C15.0–C16.9) and a cM1 status were identified in the BCR. Analyses were restricted to patients with a Belgian residence at time of diagnosis. Inclusion and exclusion criteria were in accordance with the criteria used to develop the SOURCE model [17]. As this study took place entirely within the legal framework of the Belgian Cancer Registry, no ethical approval of concerned patients was needed. We more concretely refer to the privacy law of 08/12/1992 Chapter III Art 9 §2, 2e a) and 2e b) which refers to the Health Law of 2006.

### 4.3. Procedures

Treatment type was classified as for the original SOURCE model [17]. Input parameters for initial treatment were: BSC (registered as “no treatment” or if no anti-cancer or symptom relief treatment was registered), radiotherapy (aimed at primary tumour or metastases), chemotherapy, chemoradiotherapy, chemotherapy plus short-term (≤28 days) radiotherapy, resection (aimed at primary tumour or metastases), stent placement or other treatment (all other treatments not mentioned above, like targeted therapy only).

Missing data regarding input parameters were handled using multiple imputation by chained equations [25]. Tumour staging was based on the 7th edition of the TNM staging system. However, patients diagnosed prior to 2010 were staged according to the TNM 6th edition. Conditional multiple imputation was used to align the definitions. This procedure has been described previously for SOURCE [17]. Conditional multiple imputation based on the original SOURCE dataset was also used to impute data regarding the target location (primary tumour or metastases) if patients underwent radiotherapy, since this level of detail was not given.

### 4.4. Statistical Analyses

The primary endpoint was prediction of six-month overall survival. Overall survival was defined as the time between the date of diagnosis and death, or the date of last follow-up if a patient was censored. Differences in median survival between the development and validation cohort were assessed using Cox regression. To assess model performance, a concordance index (c-index) was calculated, as well as a calibration slope, intercept, absolute error and differences between predicted and observed survival outcomes.

Model calibration was assessed by measuring the goodness-of-fit and is described by the agreement between predicted and observed outcomes. In case of a perfect prediction, the calibration line has a slope of one and an intercept of zero (x = y). A linear model was applied to assess the calibration slope and intercept of the model. The model was evaluated for the entire cohort and pre-defined patient subgroups based on the model’s input parameters [17]. Mean differences between predicted and observed survival were calculated only for patient subgroups greater than 50 patients.

A c-index was calculated to assess the discriminatory ability of SOURCE. The c-index estimate is the probability that for a random pair of patients, the patient with the highest survival indeed has a higher predicted survival estimate than the other patient. A value 0.5 indicates that the model does not perform better than chance. A value of 1 indicates perfect discrimination. C-indices <0.7 were rated as poor, 0.7–0.79 as fair, 0.8–0.89 as good and 0.9–1 as excellent [16].

The SOURCE model was re-estimated using the input of the Belgian dataset with the method that was applied to create the original Dutch SOURCE model. Model performance for the re-estimated model was also assessed by means of c-indices, calibration slopes, intercepts and absolute prediction errors.

### 4.5. Data Availability

The data that support the findings of our study are available in the Belgian Cancer Registry.

## 5. Conclusions

In conclusion, the SOURCE oesophageal model had low transportability to the Belgian population, but the gastric cancer model did transport adequately. Future studies should investigate the differences in diagnostics, treatment and survival between the populations, and the potential underlying causes. Model updating, in which newly available predictors can be incorporated to improve model performance, remains important. Furthermore, SOURCE should arm against overfitting by including fewer input parameters in future models. Lastly, for usage of the model in the Belgian clinical setting, model updating would be preferable in which ideally PS and more details regarding treatment could be incorporated.

## Figures and Tables

**Figure 1 cancers-12-00834-f001:**
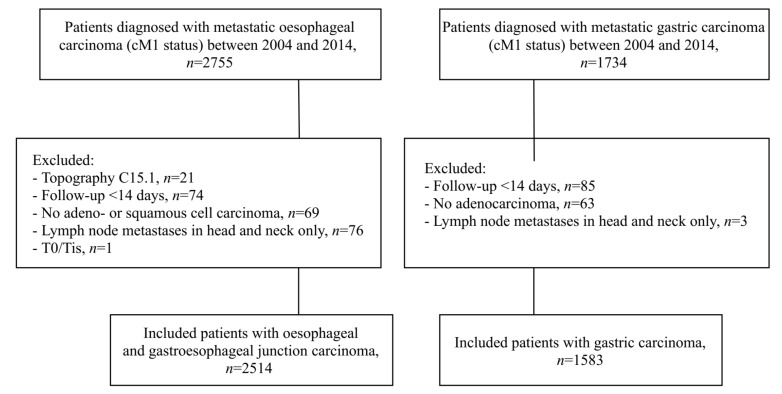
Flowchart showing inclusion of patients from the Belgian Cancer Registry in the study.

**Figure 2 cancers-12-00834-f002:**
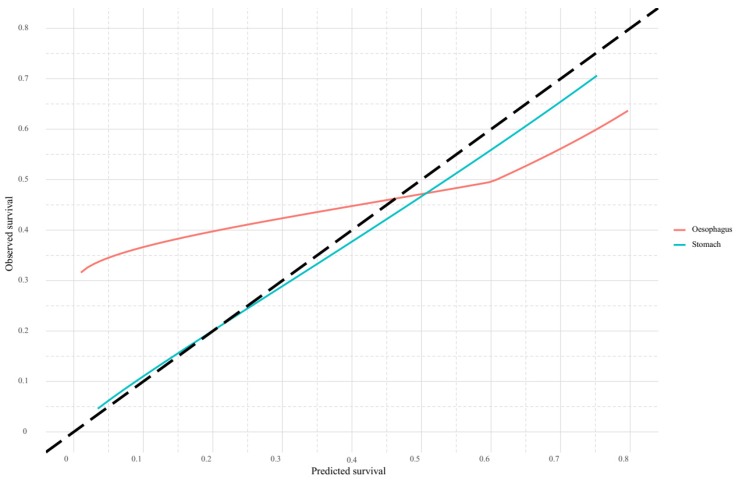
Calibration plot of predicted versus observed six-month overall survival for patients with oesophageal cancer (red line) and gastric cancer (blue line).

**Figure 3 cancers-12-00834-f003:**
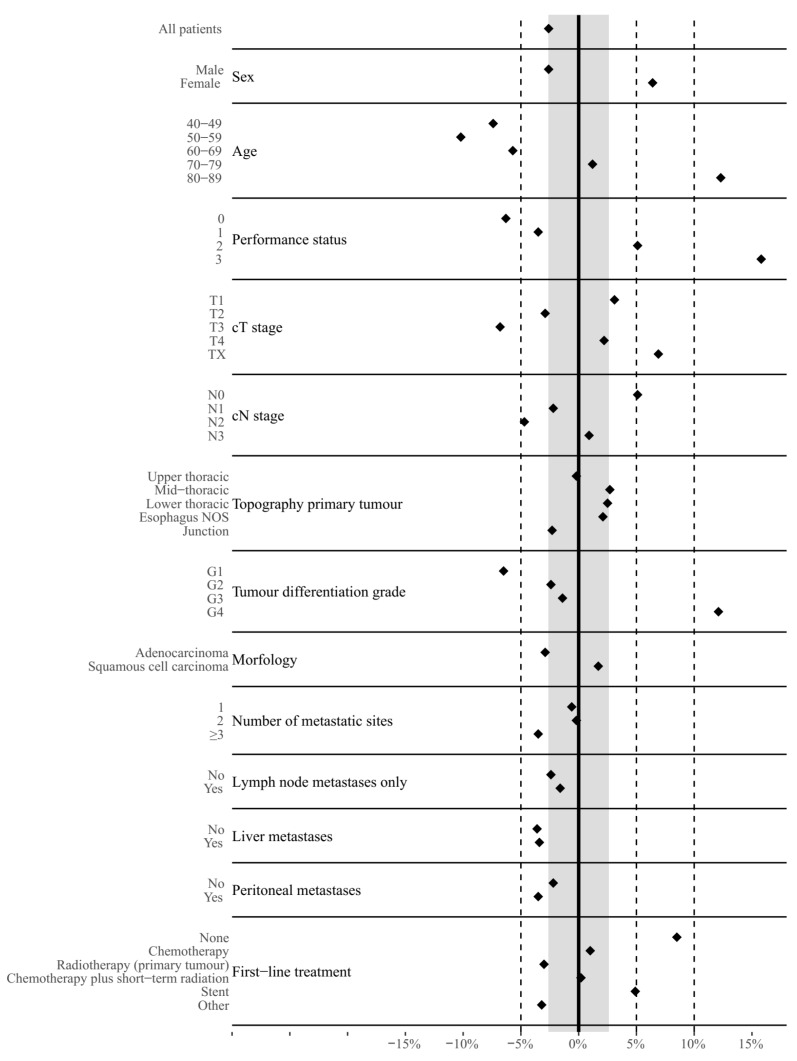
Mean differences between predicted and observed six-month overall survival for oesophageal cancer patients by patient subgroups. Values > 0% indicate an overestimation and values < 0% indicate an underestimation in overall survival. The grey band represents the mean difference between predicted and observed six-month overall survival for the entire oesophageal cancer cohort.

**Figure 4 cancers-12-00834-f004:**
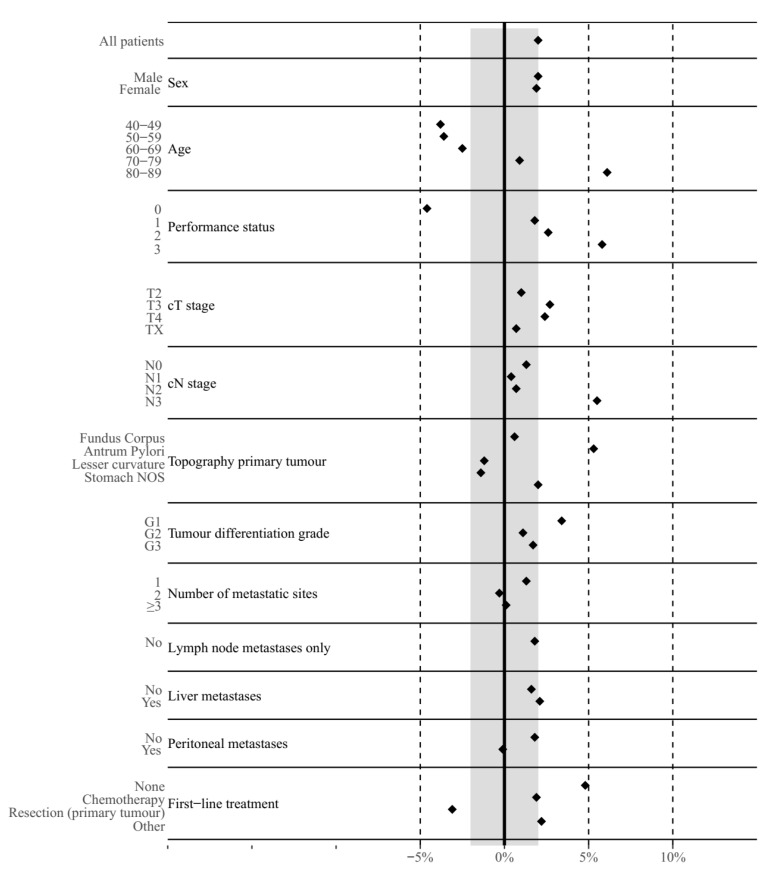
Mean differences between predicted and observed six-month overall survival for gastric cancer patients by patient subgroups. Values > 0% indicate an overestimation and values < 0% indicate an underestimation in overall survival. The grey band represents the mean difference between predicted and observed six-month overall survival for the entire gastric cancer cohort.

**Table 1 cancers-12-00834-t001:** Observed six-month overall survival of the Belgian cohort and baseline characteristics of the development (Dutch) and validation (Belgian) oesophageal cancer cohort. NOS: Not Otherwise Specified.

Patient Subgroup	Belgian Population, *n* (%)	Dutch SOURCE Population, *n* (%)	Observed Six-Month OS (%)
All patients	2514 (100)	8010 (100)	58.2
Overall survival (median (IQR) in months)	7.7 (3.2–15.3)	5.1 (2.2–10.1)	-
Sex			
Male	2031 (80.8)	6284 (78.5)	59.4
Female	483 (19.2)	1726 (21.5)	53.4
Age			
Mean (sd)	65.7 (11.6)	66.8 (10.9)	-
<40	34 (1.4)	-	-
40–49	178 (7.1)	-	76.4
50–59	566 (22.5)	-	64.6
60–69	748 (29.8)	-	64.8
70–79	680 (27)	-	51.6
80–89	292 (11.6)	-	33.2
≥90	16 (0.6)	-	-
Performance status			
Missing	248 (9.9)	-	-
0	258 (10.3)	-	71.5
1	1634 (65)	-	61.5
2	274 (10.9)	-	40.2
3	83 (3.3)	-	20.8
4	17 (0.7)	-	-
cT category			
Missing	0 (0)	1 (0)	-
T1	76 (3)	108 (1.3)	59.7
T2	269 (10.7)	1388 (17.3)	65.7
T3	1143 (45.5)	1822 (22.7)	65.1
T4	265 (10.5)	694 (8.7)	50.6
TX	760 (30.2)	3997 (49.9)	48.6
cN category			
Missing	172 (6.8)	1 (0)	-
N0	694 (27.6)	2127 (26.6)	48.7
N1	706 (28.1)	2502 (31.2)	63.2
N2	670 (26.7)	2391 (29.9)	62.7
N3	272 (10.8)	989 (12.3)	59.5
Morphology			
Adenocarcinoma	1692 (67.3)	6321 (78.9)	60.1
Squamous cell carcinoma	790 (31.4)	1423 (17.8)	55.3
Other	32 (1.3)	266 (3.3)	-
Topography primary tumour			
Cervical	15 (0.6)	44 (0.5)	-
Upper thoracic	94 (3.7)	205 (2.6)	55.3
Mid-thoracic	211 (8.4)	713 (8.9)	59.2
Lower thoracic	656 (26.1)	4461 (55.7)	58.9
Overlapping lesion	6 (0.2)	315 (3.9)	-
Junction	701 (27.9)	2112 (26.4)	62.3
Oesophagus NOS	831 (33.1)	160 (2)	53.4
Tumour differentiation grade			
Missing	373 (14.8)	3472 (43.3)	-
G1	176 (7)	112 (1.4)	58.8
G2	822 (32.7)	1464 (18.3)	59.5
G3	1051 (41.8)	2896 (36.2)	57.1
G4	92 (3.7)	66 (0.8)	57.8
Number of metastatic sites			
Missing	1058 (42.1)	267 (3.3)	-
1	457 (18.2)	4457 (55.6)	59.4
2	479 (19.1)	2208 (27.6)	59.0
≥3	520 (20.7)	1078 (13.5)	56.5
Lymph node metastases only			
Missing	1058 (42.1)	267 (3.3)	-
No	1320 (52.5)	6532 (81.5)	56.6
Yes	136 (5.4)	1211 (15.1)	70.5
Liver metastases			
Missing	1058 (42.1)	267 (3.3)	-
No	655 (26.1)	3699 (46.2)	61.5
Yes	801 (31.9)	4044 (50.5)	56.1
Peritoneal metastases			
Missing	1058 (42.1)	267 (3.3)	-
No	1293 (51.4)	7190 (89.8)	58.8
Yes	163 (6.5)	553 (6.9)	55.2
Head and neck lymph node metastases			
Missing	1058 (42.1)	267 (3.3)	-
No	1358 (54)	7232 (90.3)	-
Yes	98 (3.9)	511 (6.4)	-
Intrathoracic lymph node metastases			
Missing	1058 (42.1)	267 (3.3)	-
No	993 (39.5)	7487 (93.5)	-
Yes	463 (18.4)	256 (3.2)	-
Intra-abdominal lymph node metastases			
Missing	1058 (42.1)	267 (3.3)	-
No	990 (39.4)	6218 (77.6)	-
Yes	466 (18.5)	1525 (19)	-
First-line treatment			
None	266 (10.6)	2131 (26.6)	19.9
Chemotherapy	1247 (49.6)	2216 (27.7)	71.8
Chemotherapy plus short-term radiation	277 (11)	317 (4)	74.0
Chemoradiotherapy	45 (1.8)	80 (1)	-
Radiotherapy (primary tumour)	146 (5.8)	2081 (26)	37.7
Resection (primary tumour)	60 (2.4)	0 (0)	-
Radiotherapy (metastasis)	0 (0)	367 (4.6)	-
Resection (metastasis)	0 (0)	56 (0.7)	-
Stent	239 (9.5)	298 (3.7)	23.8
Other	234 (9.3)	464 (5.8)	51.7

**Table 2 cancers-12-00834-t002:** Calibration and discriminative ability of the entire cohort at six months survival.

Endpoint	Intercept	Slope	Absolute Error (%)	Predicted-Observed Survival (%)	Concordance Index
Oesophageal cancer model					
6-month OS	0.30 (0.28–0.31)	0.42 (0.39–0.45)	14.6 (14.5–14.7)	−2.6 (−4.3–−1.0)	0.64 (0.63–0.66)
Gastric cancer model					
6-month OS	0.02 (0.02–0.02)	0.91 (0.90–0.91)	2.5 (2.5–2.5)	2.0 (1.8–2.2)	0.66 (0.64–0.68)

**Table 3 cancers-12-00834-t003:** Observed six-month overall survival of the Belgian cohort and baseline characteristics of the development (Dutch) and validation (Belgian) gastric cancer cohort. NOS: Not Otherwise Specified.

Patient Subgroup	Belgium Population, *n* (%)	Dutch SOURCE Population, *n* (%)	Observed Six-Month OS (%)
All patients	1583 (100)	4763 (100)	46.7
Overall survival (median (IQR) in months)	5.4 (2.1–11.9)	3.9 (1.7–8.4)	-
Sex			
Male	946 (59.8)	2858 (60)	46.9
Female	637 (40.2)	1905 (40)	46.3
Age			
Mean (sd)	70 (12.8)	68.6 (12.3)	-
<40	37 (2.3)	-	-
40–49	79 (5)	-	69.6
50–59	185 (11.7)	-	60.0
60–69	375 (23.7)	-	56.8
70–79	514 (32.5)	-	44.8
80–89	363 (22.9)	-	27.0
≥90	30 (1.9)	-	-
Performance status			
Missing	131 (8.3)	-	-
0	143 (9)	-	67.3
1	944 (59.6)	-	50.4
2	255 (16.1)	-	34.4
3	79 (5)	-	17.1
4	31 (2)	-	-
cT category			
Missing	145 (9.2)	1 (0)	-
T1	45 (2.8)	58 (1.2)	-
T2	119 (7.5)	659 (13.8)	52.8
T3	374 (23.6)	672 (14.1)	51.5
T4	257 (16.2)	802 (16.8)	44.3
TX	643 (40.6)	2571 (54)	43.4
cN category			
Missing	141 (8.9)	0 (0)	-
N0	767 (48.5)	2366 (49.7)	44.6
N1	259 (16.4)	1012 (21.2)	48.1
N2	336 (21.2)	1264 (26.5)	49.4
N3	80 (5.1)	121 (2.5)	50.9
Topography primary tumour			
Fundus	105 (6.6)	162 (3.4)	54.4
Corpus	158 (10)	954 (20)	47.5
Antrum Pylori	238 (15)	1075 (22.6)	49.6
Pylorus	26 (1.6)	239 (5)	-
Lesser curvature NOS	63 (4)	181 (3.8)	50.8
Greater curvature NOS	34 (2.1)	106 (2.2)	-
Overlapping lesion	8 (0.5)	1645 (34.5)	-
Stomach NOS	951 (60.1)	401 (8.4)	44.6
Tumour differentiation grade			
Missing	285 (18)	2180 (45.8)	-
G1	101 (6.4)	42 (0.9)	48.5
G2	314 (19.8)	488 (10.2)	46.3
G3	847 (53.5)	2028 (42.6)	46.7
G4	36 (2.3)	25 (0.5)	-
Number of metastatic sites			
Missing	658 (41.6)	168 (3.5)	-
1	355 (22.4)	3099 (65.1)	46.0
2	273 (17.2)	1067 (22.4)	45.1
≥3	297 (18.8)	429 (9)	48.9
Lymph node metastases only			
Missing	658 (41.6)	168 (3.5)	-
No	897 (56.7)	4141 (86.9)	46.6
Yes	28 (1.8)	454 (9.5)	-
Liver metastases			
Missing	658 (41.6)	168 (3.5)	-
No	463 (29.2)	2873 (60.3)	50.8
Yes	462 (29.2)	1722 (36.2)	42.3
Peritoneal metastases			
Missing	658 (41.6)	168 (3.5)	-
No	459 (29)	2735 (57.4)	45.5
Yes	466 (29.4)	1860 (39.1)	48.0
Head and neck lymph node metastasis			
Missing	658 (41.6)	168 (3.5)	-
No	906 (57.2)	4538 (95.3)	-
Yes	19 (1.2)	57 (1.2)	-
Intrathoracic lymph node metastasis			
Missing	658 (41.6)	168 (3.5)	-
No	839 (53)	4419 (92.8)	-
Yes	86 (5.4)	176 (3.7)	-
Intra-abdominal lymph node metastasis			
Missing	658 (41.6)	168 (3.5)	-
No	642 (40.6)	3973 (83.4)	-
Yes	283 (17.9)	622 (13.1)	-
First-line treatment			
None	531 (33.5)	2266 (47.6)	16.8
Chemotherapy	825 (52.1)	1648 (34.6)	64.5
Chemotherapy plus short-term radiation	28 (1.8)	52 (1.1)	-
Chemoradiotherapy	0 (0)	0 (0)	-
Radiotherapy (primary tumour)	29 (1.8)	154 (3.2)	-
Resection (primary tumour)	111 (7)	247 (5.2)	63.1
Radiotherapy (metastasis)	0 (0)	63 (1.3)	-
Resection (metastasis)	0 (0)	97 (2)	-
Stent	0 (0)	56 (1.2)	-
Other	59 (3.7)	180 (3.8)	45.8

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
