# Peer review of "External Validation of the Dutch SOURCE Survival Prediction Model in Belgian Metastatic Oesophageal and Gastric Cancer Patients"

_cancers, 2020, doi:10.3390/cancers12040834_

Round 1

Reviewer 1 Report

Dear Authors of the manuscript entitled: External validation of the Dutch SOURCE survival prediction model in Belgian metastatic oesophageal and gastric cancer patients,

Thank you for submitting your work to Cancers. The study is well conducted and the manuscript well constructed. Perhaps the high percentage of missing data in oesophageal cancer patients (number of metastatic sites, lymph node metastases, peritoneal metastases, liver metastases etc.) reaching over 40% is the reason for unsatisfactory results. However, this is not directly related to the prediction model itself, therefore I recommend this paper for publication.

Introduction: general summary of the clinical problem has been described accordingly.

Results: clinical examination for evaluation of the prediction model was described in a well manner.

Discussion: All statements made were supported by evidence.limitations of the study (poor fit in the oesophageal cancer) were precisely highlighted.

Materials and Methods: As mentioned by Authors, the manuscript was written in accordance with TRIPOD statement.

Author Response

Dear Reviewer,

Thank you for your comments and suggestions.

We agree that the relatively high percentage of missing data regarding the location of metastases (>40%) could possibly explain the results observed for the oesophageal model. Although we have used the same imputation method for these data as was used during model development, this might still influence model performance during validation. In addition, the oesophageal model contains more input parameters based on the location of metastases. So both factors could contribute to the somewhat unsatisfactory results, as stated on page 10, line 168-173.
We agree that these missing data is not related to the prediction model itself, since these data were available in the Dutch registry during model development.

Reviewer 2 Report

Your study provides a validation of the SOURCE model in an external population. It shows that the model provides a good fit for stomach, but not esophageal cancers.

I suggest the following changes to improve the manuscript:

- In the methods section (not the supplementary) you need to describe in detail how the models for esophageal and gastric tumors are exactly applied.

Figure S1 should be moved to the main text.

- One major limitation of the models is that, as you claim, they should be a basis for discussing active treatments versus BSC with patients. However, they provide survival estimates based on a population which mostly received active treatment. Therefore, an "unbiases" prognosis of expected survival is not possible. This issue merits extensive discussion.

- Do you think it is justified to combine esophageal squamous cell and adenocarcinoma into one prediction model? In my view, these are biologically rather different tumor entities. For many treatments, esophageal adenocarcinoma is rather combined with gastric adenocarcinoma than esophageal squamous cell carcinoma. This might explain the relatively poor fit of the esophageal cancer model. You need to discuss this issue.

Author Response

Author's Reply to the Review Report (Reviewer 2)

Dear Reviewer,

Thank you for your constructive comments, suggestions and questions. Hereby, we present a detailed point-by-point reply:

Figure S1 should be moved to the main text.

We have moved Supplementary Figure 1 to the main text. We understand that providing these data for oesophageal and gastric cancer patients provides a better overview and understanding for the reader. We would like to ask the Editorial Board of Cancers to place the additional Figure (now named Figure 3) at the right page and place in the manuscript, according to your lay-out guidelines. For now, the figure is placed on page 7, line 137-141.

In the methods section (not the supplementary) you need to describe in detail how the models for esophageal and gastric tumors are exactly applied.

Thank you for your suggestion to improve the manuscript with information on application in clinical practise. The following paragraphs are moved from the supplementary methods to the methods in the main manuscript:

“The SOURCE model aims to stimulate evidence based, personalized and tailored information provision to improve decision making after oesophageal-gastric cancer diagnosis. The model predicts overall survival for patients with metastatic oesophageal or gastric carcinoma (cM1), who did not die within 14 days after diagnosis. Patients with only distant metastases located in the head or neck region fall outside the target population of SOURCE.

Input parameters of the model include: age, cT-category, cN-category, tumour differentiation grade, number of metastatic sites, distant lymph node metastasis only, intra-thoracic and intra-abdominal lymph node metastasis, and initial treatment. The gastric cancer model also includes gender as an input parameter and the oesophageal cancer model also includes peritoneal, liver and head and neck metastases, morphology, and topography. Input parameters were measured at diagnosis, before the start of treatment.“ (page 11, line-225-235).

In addition the following paragraph is added to describe the application of the model in practise: ‘’SOURCE is integrated into a web-interface and will be made freely available after extensive assessments in clinical practice. Physicians can use the model together with patients during the clinical consultation. Since medical terminology is present in the web-interface, it is recommended that physicians discuss the results from the model with the patient, in a way that is tailored to the patient's level of understanding. It should be noted that SOURCE is developed to be a decision-aid to stimulate shared- and informed decision making. It should not and cannot replace the expertise and clinical judgement of physicians’’. (page 12, line-236-424).

One major limitation of the models is that, as you claim, they should be a basis for discussing active treatments versus BSC with patients. However, they provide survival estimates based on a population which mostly received active treatment. Therefore, an "unbiases" prognosis of expected survival is not possible. This issue merits extensive discussion.

We agree that survival estimation could be biased when predictions are made for relatively fit patients, because of likely selection bias. The original development paper discussed this issue as well, since it often occurs in population-based analysis. Since the sample size of patients on BSC in the development sample was substantial (26.6% and 47.6% for oesophageal and gastric cancer respectively), we believe that survival estimation can be performed. However, this still does not address the issue of selection bias in the survival predictions for patients who are candidates for systemic treatment or other forms of active treatment. Therefore we added the following paragraph to strengthen our paper:

“Lastly, SOURCE was developed to aid decision making between BSC and (some form of) active treatment. During model development, 26.6% (n=2,131) and 47.6% (n=2,266) of Dutch oesophageal and gastric cancer patients received BSC (no treatment). Although this relatively large cohort could aid survival estimation on BSC, it should be pointed out that these estimates may have an inherent selection bias. Patients who received BSC most likely had worse PS scores or comorbidities compared to patients who did undergo treatment. Therefore survival estimates for a relatively fit patient considering BSC may be underestimated. Although this effect could be partially corrected by other input parameters in the model, there may still be bias in the survival predictions.” (page 11, line 215-222).

Do you think it is justified to combine esophageal squamous cell and adenocarcinoma into one prediction model? In my view, these are biologically rather different tumor entities. For many treatments, esophageal adenocarcinoma is rather combined with gastric adenocarcinoma than esophageal squamous cell carcinoma. This might explain the relatively poor fit of the esophageal cancer model. You need to discuss this issue.

We understand the controversy between the grouping based on topographgy (oesophagus versus stomach) and tumour histology (adenocarcinoma versus squamous cell carcinoma). The current combination is based on topography. This category can be criticized for combining biologically different tumor entities. An attempt to correct for survival differences based on histology was done by including it as an input parameter of the oesophageal model. It is unknown whether this could explain the relatively poor fit of the esophageal cancer model. Our validation study was bound to this topographic grouping, since it was chosen in the original development model, and is thus anchored in the underlying two models developed earlier.

In the manuscript, we have added the following paragraph:

“Furthermore, adenocarcinoma and squamous cell carcinoma were combined into the same oesophageal cancer model, despite their differential biological features. Although the oesophageal cancer model contained histology as an input parameter, it is unclear to what extent this combination contributes to the poor model fit. Patient subgroup analysis showed that mean differences between predicted and observed survival for adenocarcinoma, squamous cell carcinoma, and the entire cohort were -2.9%, +1.7%, and -2,6%, respectively. These mean differences do not substantially differ (see Figure 3). (page 10-11, line-183-189).

Round 2

Reviewer 2 Report

Thank you for addressing all the issues I raised in my initial review.